# Beyond Metabolism: Psychiatric and Social Dimensions in Bariatric Surgery Candidates with a BMI ≥ 50—A Prospective Cohort Study

**DOI:** 10.3390/nu17152573

**Published:** 2025-08-07

**Authors:** Marta Herstowska, Karolina Myśliwiec, Marta Bandura, Jędrzej Chrzanowski, Jacek Burzyński, Arkadiusz Michalak, Agnieszka Lejk, Izabela Karamon, Wojciech Fendler, Łukasz Kaska

**Affiliations:** 1Department of Pediatrics, Diabetology and Endocrinology, Medical University of Gdansk, 80-210 Gdansk, Poland; karolina.mysliwiec@autonomik.pl (K.M.); ma.bromirska@gmail.com (M.B.); agnieszka.lejk@op.pl (A.L.); 2Department of Biostatistics and Translational Medicine, Medical University of Lodz, 92-215 Lodz, Poland; jedrzej.chrzanowski@umed.lodz.pl (J.C.); arkadiusz.michalak@umed.lodz.pl (A.M.); wojciech.fendler@umed.lodz.pl (W.F.); 3Department of Dermatology, Venereology and Allergology, Medical University of Gdansk, 80-210 Gdansk, Poland; izabela.blazewicz@gumed.edu.pl; 4Hospital of the Ministry of the Interior and Administration, 80-104 Gdansk, Poland; lukasz.kaska@wp.pl

**Keywords:** obesity, bariatric surgery, psychiatric dimension

## Abstract

**Background:** Super morbid obesity (SMO), defined as a body mass index (BMI) ≥ 50 kg/m^2^, represents a distinct and increasingly prevalent subgroup of patients undergoing bariatric surgery. Compared to individuals with lower BMI, patients with BMI ≥ 50 kg/m^2^ often exhibit unique clinical, psychological, and social characteristics that may influence treatment outcomes. **Objective:** This study aimed to compare demographic, metabolic, and psychiatric profiles of patients with BMI ≥ 50 kg/m^2^ and non-super morbid obesity (NSMO; BMI < 50 kg/m^2^) who were evaluated prior to bariatric surgery. **Methods:** A total of 319 patients were recruited between December 2022 and December 2023 at a bariatric center in Gdansk, Poland. All participants underwent a comprehensive preoperative assessment, including laboratory testing, psychometric screening (BDI, PHQ-9), and psychiatric interviews. Patients were stratified into class IV obesity and NSMO groups for comparative analysis. **Results:** Patients with BMI ≥ 50 kg/m^2^ were significantly older and more likely to report a history of lifelong obesity, family history of obesity, and childhood trauma. They had higher rates of obesity-related health problems such as hypertension, obstructive sleep apnea, and chronic venous insufficiency, as well as worse liver function and lipid profiles. Although the overall psychiatric burden was high in both groups, patients with BMI ≥ 50 kg/m^2^ reported fewer prior diagnoses of depression and eating disorders, despite similar scores on screening tools. **Conclusions:** Patients with BMI ≥ 50 kg/m^2^ represent a clinically distinct population with elevated metabolic risk, complex psychosocial backgrounds, and possibly underrecognized psychiatric burden. These findings underscore the need for multidisciplinary preoperative assessment and individualized treatment strategies in this group of patients.

## 1. Introduction

Obesity is a chronic disease with multifactorial etiopathogenesis, involving genetic, metabolic, environmental, lifestyle, and mental factors [1]. It is associated with metabolic disorders, cardiovascular complications, musculoskeletal diseases, cancer, mental health disorders, and numerous other comorbidities [2]. According to the World Health Organization (WHO) guidelines, obesity is defined using the body mass index (BMI), with a BMI ≥ 30 considered diagnostic. It is further classified into three main categories: class I (BMI 30.0–34.9), class II (BMI 35.0–39.9), and class III (BMI ≥ 40.0) [3]. Some sources further distinguish class IV (BMI ≥ 50.0) and class V (BMI ≥ 60.0), commonly referred to as “super obesity” (SO) or “super morbid obesity” (SMO). Evidence from the literature suggests that this subgroup may differ significantly from individuals with lower BMI, both in social and psychological characteristics and in the effectiveness of bariatric treatments [4].

Over the past five decades, the prevalence of obesity has increased dramatically, reaching pandemic proportions. According to a 2022 report by Poland’s Supreme Audit Office, an estimated nine million adults in Poland are affected by obesity [5]. However, most sources emphasize that these figures are likely underestimated due to the lack of a comprehensive patient registry. In parallel, increasing attention is being directed toward the relationship between obesity and mental health, as obesity serves as a risk factor for various conditions that negatively affect both psychological well-being and health-related quality of life [6]. Despite significant scientific advancements in the development of pharmacological treatments, including GLP-1 analogs (liraglutide, semaglutide), orlistat, and naltrexone–bupropion [7,8], bariatric surgery remains the only method with proven long-term efficacy in patients with BMI ≥ 50 kg/m^2^. Surgical treatment in this population should be performed by a multidisciplinary team to ensure thorough pre- and postoperative care, which is essential for optimal long-term outcomes. Observational cohort studies are therefore needed to investigate the most common challenges in this group, identify clinically distinct subgroups, and trace their associations with patients’ prognosis.

In this study, we aimed to analyze and characterize a group of patients with BMI ≥ 50 and compare them with those with non-super morbid obesity (NSMO; BMI < 50), all evaluated prior to bariatric surgery. We hope to contribute to a better understanding of these patients and aid in selecting optimal diagnostic and therapeutic approaches.

## 2. Materials and Methods

Patients were recruited at the Hospital of the Ministry of the Interior and Administration in Gdansk between December 2022 and December 2023. The study was reviewed and approved by the Bioethics Committee for Scientific Research at the Medical University of Gdansk (Project Number: NKBBN/868/2021).

As part of the bariatric qualification process, patients were initially assessed by a bariatric surgeon to determine eligibility for surgical treatment. Those qualified underwent standard preoperative procedures, including multiple specialist consultations (dietitian, psychologist, psychiatrist, cardiologist) and comprehensive diagnostic investigations. During the psychiatric consultation, patients were informed about the study and provided written informed consent to participate. They were also asked to complete study-specific psychometric scales and questionnaires.

Included participants underwent the following procedures: fasting blood samples were collected for laboratory diagnostics, including glucose, insulin, C-reactive protein (CRP), glycated hemoglobin (HbA1c), lipid profile, and liver function parameters. Anthropometric measurements were also performed, including the calculation of the patients’ BMI. BMI was calculated based on the highest recorded body weight in the patient’s medical documentation. This approach was chosen because patients with fluctuating body weight may have earlier measurements that do not accurately reflect their true obesity level at the time of study inclusion. Moreover, some patients intentionally reduced their weight before their first visit to the bariatric surgeon, which could have led to an underestimation of their pre-treatment condition.

Additionally, patients were asked to complete psychometric scales assessing the severity of depressive symptoms, including the Beck Depression Inventory (BDI), which is a 21-item self-report questionnaire designed to measure the intensity of depressive symptoms. Each item is rated from 0 to 3, and the total score ranges from 0 to 63; higher scores indicate more severe depression. The Patient Health Questionnaire-9 (PHQ-9), a 9-item scale based on DSM criteria, was also administered to assess the frequency of depressive symptoms over the previous two weeks, with total scores ranging from 0 to 27, where higher values similarly reflect greater symptom severity. In addition, as part of the psychiatric assessment, information was obtained about patients’ experiences of childhood abuse, defined as physical, emotional, and/or neglect-related maltreatment.

Patients were also asked to complete a proprietary questionnaire that collected demographic data, assessed social and cognitive functioning, and included a medical history interview focusing on chronic somatic conditions. This questionnaire also included questions regarding subjective sleep quality.

### Statistical Analysis

Patients were classified into either group with BMI ≥ 50 kg/m^2^ or the group with BMI < 50 kg/m^2^. Comparisons between the groups of continuous variables were performed using the Mann–Whitney U-test (due to lack of normality, as tested by the Shapiro–Wilk test) and reported using median with lower and upper quartile. Dichotomous variables were compared using Chi^2^ or other appropriate test and reported with odds ratio (OR) and 95% confidence interval (95% CI). Alpha was set at 0.05, and no correction for multiple testing was applied. All statistical procedures were performed using STATISTICA software (v13.3, TIBCO software Inc., Palo Alto, CA, USA) and RStudio (version 4.3.3).

## 3. Results

### 3.1. Group Characteristics

A total of 376 patients qualified for bariatric surgery, and 366 began the preparatory process for the procedure. Twenty-six patients did not meet the criteria for surgery or were deferred. Ultimately, 340 patients were included in the study, and we obtained 319 complete records (Figure 1).

In total, N = 340 patients met the inclusion criteria and were recruited into the study. Out of those, full data was collected from N = 319, comprising 72 (22.57%) patients with super morbid obesity (SMO) and 247 (77.43%) with non-super morbid obesity (NSMO). Patients with SMO did not differ significantly in height or sex compared to those with NSMO, but were significantly older (median [Q1–Q3]: 41.00 [34.00–50.00] vs. 34.00 [29.00–43.00], *p* < 0.001).

We did not observe any significant differences in unemployment rate (OR: 1.46 95% CI: 0.80–2.62) between the study groups. Interestingly, while patients with SMO did not show a significant difference in attaining higher education (OR: 1.26 95% CI: 0.71–2.23), they were more likely to come from families without a history of higher education (OR 0.41, 95% CI 0.21–0.78). Moreover, patients with BMI ≥ 50 kg/m^2^ more frequently reported lifelong obesity (OR: 4.58, 95% CI: 2.45–8.92), had a higher prevalence of positive family history of obesity (OR: 9.78, 95% CI: 1.57–405.33) and more often reported being victim of violence against children (OR: 3.73, 95% CI: 2.07–6.76).

The vast majority of participants were in a relationship at the time of the study, with no significant difference between patients with BMI ≥ 50 kg/m^2^ and patients with BMI < 50 kg/m^2^ in overall relationship status (OR: 3.36 95% CI: 1.82–6.17). However, patients with BMI ≥ 50 kg/m^2^ were significantly more likely to be in formal relationships compared to those with NSMO (OR: 3.36, 95% CI: 1.83–6.17).

Detailed group characteristics, along with a summary of socioeconomic data, are presented in Table 1 and Table 2.

### 3.2. Comorbidities and Laboratory Examination

Patients with BMI ≥ 50 kg/m^2^ more often reported obesity-related health problems such as hypertension (OR: 3.52, 95% CI 1.85–7.07), obstructive sleep apnoea (OR: 23.67, 95% CI: 6.06–203.12), and chronic venous insufficiency (OR: 104.11, 95% CI: 34.34–428.85). No differences were observed in the frequency of stroke or heart attack history, fatty liver disease, or type 2 diabetes mellitus (DM2) between the study groups. However, patients with BMI ≥ 50 kg/m^2^ did exhibit significantly higher concentrations of AST and ALT, as well as elevated levels of total cholesterol and triglycerides, suggesting poorer metabolic and hepatic control in this group. We did not observe significant differences in HbA1c and fasting glucose levels between study groups, although patients with BMI ≥ 50 kg/m^2^ had significantly higher insulin and HOMA-IR levels compared with the NSMO group. Detailed statistics of comorbidities and laboratory results are presented in Table 3 and Table 4.

### 3.3. Mental Health and Psychiatric History

A psychiatric examination was conducted to assess the current psychiatric burden in our cohort. BDI and PHQ9 scales were used as screening tools for depressive disorders. Information on psychiatric history, sleep quality, and related conditions is presented in Table 5 and Figure 2.

BDI—Beck Depression Inventory, PHQ9—Patient Health Questionnaire 9, SMO—Super-Morbid Obesity, NSMO—Non-Super Morbid Obesity

At the time of the study, 67 (21.00%) patients were receiving psychiatric care. A total of 136 (42.63%) patients reported a prior psychiatric diagnosis, including 112 (35.11%) with depressive disorders. As expected, patients with depressive disorders had significantly higher BDI and PHQ9 scores (Figure 2A,B). Among patients without a prior diagnosis of depression, 18 (10.17%) were classified as having severe depression according to the BDI, and 12 (6.78%) exhibited symptoms of severe depression according to the PHQ9. All of these patients received a clinical diagnosis of depression following their psychiatric consultation.

Patients with SMO did not differ significantly from those with NSMO in terms of BDI (Figure 2C) and PHQ9 (Figure 2D) scores, nor in the frequency of current psychiatric treatment (OR: 0.54, 95% CI: 0.23–1.15). However, patients with SMO reported depressive disorders significantly less frequently than those with NSMO (OR: 0.28, 95% CI: 0.14–0.52). Surprisingly, patients with BMI ≥ 50 kg/m^2^ also reported binge eating disorder (BED) significantly less frequently compared to those with NSMO (OR: 0.31, 95% CI: 0.17–0.55). Moreover, patients with NSMO were significantly more likely to have at least one eating disorder compared to patients with BMI ≥ 50 kg/m^2^ (OR: 0.29, 95% CI: 0.16–0.50, *p* < 0.001).

In terms of sleep habits, patients with BMI ≥ 50 kg/m^2^ more frequently reported poor sleep quality (OR: 2.14, 95% CI: 1.19–3.95) and snoring (OR: 11.95, 95% CI: 4.95–35.02), reflecting a higher occurrence of obstructive sleep apnoea in this group. Conversely, patients with SMO reported using hypnotic drugs less frequently than those with NSMO (OR: 0.13, 95% CI: 0.00–0.84). No significant differences were observed in prolonged sleep latency, nocturnal awakenings, or early morning awakenings.

## 4. Discussion

Numerous studies suggest a strong correlation between patients with BMI ≥ 50 kg/m^2^ and its impact on psychosocial functioning, professional life, and the risk of developing depression and eating disorders [9]. In this group, obesity has typically been present since early childhood, with a higher prevalence of positive family history of the condition. These patients often have no experience of maintaining a normal body weight (76% of patients with BMI ≥ 50 kg/m^2^ in our cohort), frequently due to unhealthy eating patterns established within their family environment [10].

In our study, we did not observe significant differences between the patients with BMI ≥ 50 kg/m^2^ and NSMO groups in terms of education or unemployment. However, Jean O’Connell et al. reported higher unemployment among individuals with a BMI ≥ 50 kg/m^2^, suggesting that the relationship between severe obesity and employment status may vary depending on the population studied. With reference to the same study, individuals with a BMI ≥ 50 were less likely to be married (51%), compared to 72% in the group with a BMI < 50. Our findings are consistent with this, with 55,6% of patients with BMI ≥ 50 kg/m^2^ being in formal relationships/marriage.

Interestingly, NSMO patients in our cohort more frequently reported being in informal partnerships (51.01%), which may reflect age-related differences between populations, as the mean age of NSMO patients in our study was 34 years, compared to 50 (SD 15) and 46 (SD 13) in the BMI 30–39 and 40–49 groups, respectively, in the cited study [11].

Beyond socioeconomic and relational factors, patients with BMI ≥ 50 kg/m^2^ were significantly more likely to report a history of violence (50%) compared to those with NSMO (21.05%). This finding is consistent with the study by Ryan J. Marek et al., which demonstrated a higher prevalence of sexual abuse among patients with BMI ≥ 60 kg/m^2^ [12]. The strong association between severe obesity and exposure to trauma highlights the need to consider psychological and psychiatric dimensions when assessing this population. Obesity has been linked to various mental health disorders, including depression, anxiety, and stress-related conditions. It may exacerbate negative self-perception, often reinforced by social stigma, while atypical depression can lead to compulsive eating as a maladaptive coping mechanism. Additionally, stress-induced overeating, driven by emotional dysregulation, may contribute to a self-perpetuating cycle in which psychological distress and obesity mutually reinforce one another [13].

Milaneschi et al. identified several common mechanisms between depression and obesity, including chronic inflammation, stress, dysregulation of the hypothalamic–pituitary–adrenal axis, abnormalities in the regulation of insulin and leptin levels, as well as shared genetic and socioeconomic factors [14]. Our study supports these findings, with over 40% of our study group receiving a psychiatric diagnosis, including depression. Similar results were observed in the study by Jiwanmall et al., where over 30% of patients with morbid obesity were diagnosed with depression [15]. However, another publication reported different results, with only 22% of patients with obesity showing depressive symptoms, possibly due to the use of different diagnostic criteria, as the study relied on a proprietary depression scale rather than standardized instruments or the Diagnostic and Statistical Manual of Mental Disorders (DSM-V) criteria [16].

Our observations reveal an unexpected finding: a lower percentage of SMO patients with a history of psychiatric treatment (20.83%) and a significantly lower rate of depressive episode diagnoses in this group (18.06% vs. 40.08%). The diagnosis of a depressive episode was based on medical history interviews; however, during the psychiatric consultations conducted as part of this study, some patients from both groups received a new diagnosis of depression. Subjective mood assessments using psychometric scales confirm the presence of mood disorders; however, these do not correlate with the need for psychiatric treatment. Analysis of clinical interviews suggests that the underestimation of depression in patients with BMI ≥ 50 kg/m^2^ may be linked to passive-dependent personality traits, behavioral inhibition, lack of experiences related to an improved quality of life due to years of obesity, and the patients’ self-perception of being ineffective [17].

The literature reports varying prevalence rates of eating disorders in the SMO patient group, with the vast majority of those qualified for bariatric surgery struggling with disordered eating patterns [12]. According to the results presented by Arianna Belloli et al., 92.7% of bariatric surgery candidates exhibited two or more pathological eating behaviors, while 31.9% (n = 30) of them showed moderate to severe binge eating, as measured by the Binge Eating Scale [18]. In our study, no significant differences were observed in the frequency of bulimia and NES. The findings regarding the less frequent diagnosis of BED in the SMO patient group remain insufficiently understood. Moreover, the NSMO group exhibited a higher prevalence of at least one eating disorder compared to the SMO group. However, when analyzing the available literature [19] and drawing on our own experiences, it appears that patients with BMI ≥ 50 kg/m^2^, due to a persistent focus on food and intense positive emotions related to achieving gratification, engage in continuous eating. In the BMI < 50 group, patients seem to be more motivated during the preparatory process for bariatric surgery and are more likely to adopt a regular, calorie-reducing eating pattern. Nevertheless, they frequently lose control over eating, which clinically manifests as binge eating episodes.

It should be noted that patients with BMI ≥ 50 kg/m^2^ are at a higher risk of developing serious chronic diseases. Our study supports this observation, highlighting a greater prevalence of hypertension, obstructive sleep apnoea and chronic venous insufficiency. However, our findings differ from those of recent studies, which did not report such a strong association between high levels of obesity and the incidence of hypertension [12]. When analyzing the results of obesity-related health problems occurrences, our observations are consistent with Stahel, Sud et al. study, which indicates a higher rate of sleep apnoea without significant differences in the frequency of heart diseases, in the group of SMO patients [4]. Myocardial infarction and stroke were rarely observed in our cohort, likely due to the relatively young age of the patients. In contrast, hepatic steatosis was commonly reported and prevalent in both groups. The occurrence of sleep apnoea in 97.22% of patients with SMO clinically affects the poorer quality of sleep, disturbances in the sleep initiation and promotion phases. However, it is interesting to observe that sleep medications are used much less frequently in this group of patients.

Among the laboratory parameters in the SMO group, liver function tests, lipid profile, insulin levels, and CRP are significantly higher. Elevated inflammatory markers correlate with the diagnosis of sleep apnoea, insulin resistance, and are a risk factor for the development of depression in patients with obesity [19]. The lack of significant differences in glucose and HbA1c values compared to NSMO patients is surprising; however, the self-reported prevalence of diabetes was identical in both groups. This observation may be attributed to a limitation of our study, namely the lack of data on antidiabetic medication use.

## 5. Conclusions

The unique aspect of our study lies in the multidimensional assessment of a relatively large group of patients with BMI ≥ 50 kg/m^2^ with a focus on demographic, metabolic, and psychiatric factors that warrant special attention in the bariatric process, particularly in the context of long-term treatment outcomes. Given the distinct characteristics of this population and the specific selection criteria, we emphasize the importance of individualizing bariatric treatment, particularly in recognizing mental health disorders. While data on the impact of bariatric surgery in this group are still being analyzed, our study serves as a starting point, highlighting the need for further research to better understand these patients and optimize therapeutic strategies.

## 6. Limitations

The main limitation of the study was to attempt to create relatively homogeneous groups for comparison. We did not differentiate based on age, comorbidities, etc., which was due to the relatively small initial number of patients with BMI ≥ 50 kg/m^2^. We were concerned that introducing further subdivisions within the group would reduce the sample size and representativeness.

Another limitation of the study is the analysis of some variables based on non-standardized questionnaires. The ad hoc questionnaires were developed based on observations of therapists and physicians participating in follow-up visits, who often obtained information affecting patient functioning. This information, which could not be captured by well-known and widely used psychometric scales, would prevent the patient’s condition, both pre- and post-operatively, from being captured. Expanding our patient interviews allowed us to gather a wealth of information that not only personalized the process for each patient but, we hope, will contribute to a long-term focus on factors that may indicate even better long-term outcomes and focus on factors associated with the superior success of the bariatric process.

This is a multifaceted assessment, but the baseline assessment of patients with BMI ≥ 50 kg/m^2^ is much more valuable. Follow-up of this group after bariatric treatment is much more valuable, and we expect to confirm the interesting observations that are being developed one and two years after the procedure. What seems valuable in this study is the comprehensive analysis of the characteristics of patients with BMI ≥ 50 kg/m^2^.

Our preliminary observations indicate that even though there are limitations to our study, such as a single-center study, retrospective self-reporting, some psychiatric conditions were diagnosed post hoc during the study, or even incomplete data from the study group, we are convinced of its innovation.

## Figures and Tables

**Figure 1 nutrients-17-02573-f001:**
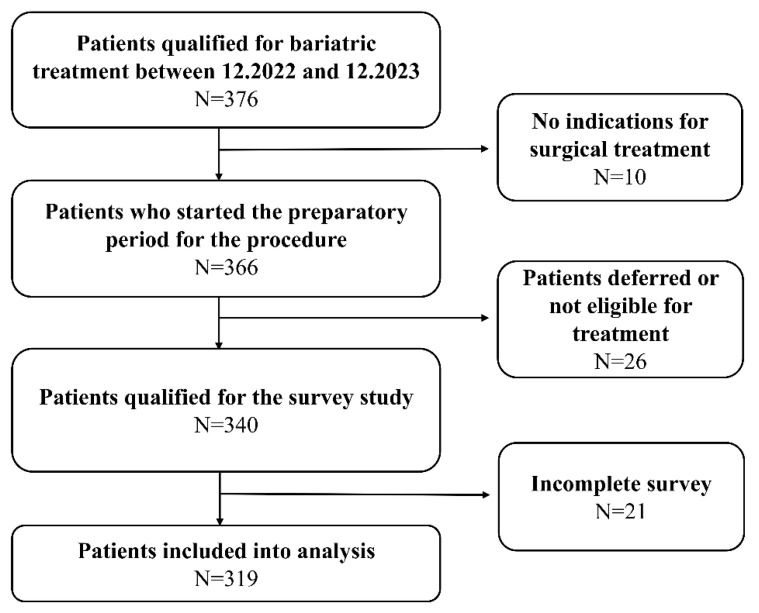
Summary of patient recruitment and data collection process.

**Figure 2 nutrients-17-02573-f002:**
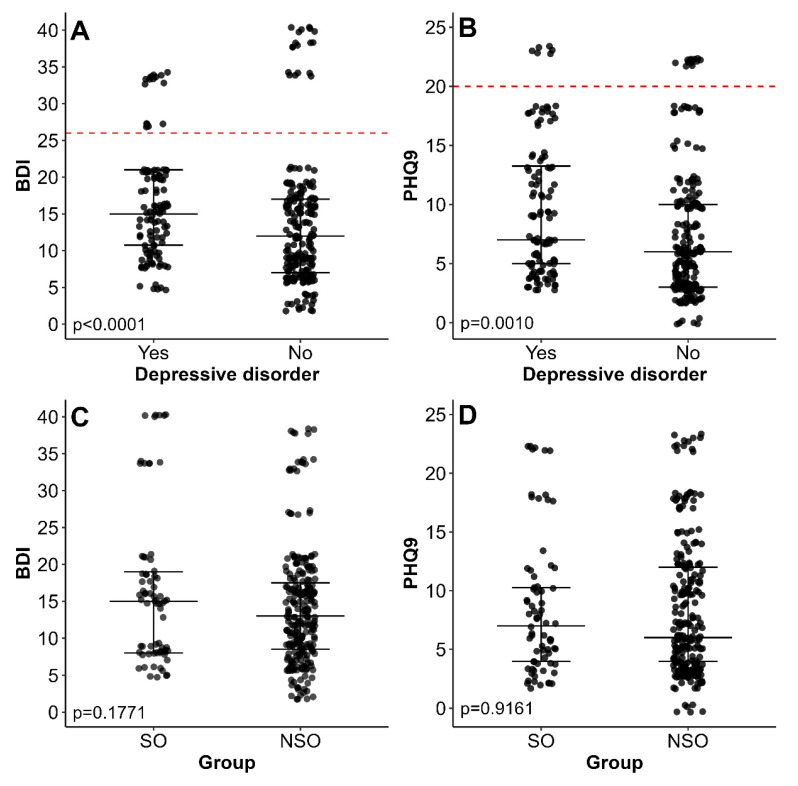
Beck Depression Inventory (BDI) and Patient Health Questionnaire (PHQ-9) scores characteristics regarding prior depression diagnosis (panels (**A**,**B**)) and presence of Super-Morbid Obesity (SMO, panels (**C**,**D**)). The middle line denotes the median, while the whisker span 25–75% range. The horizontal, red lines on panels (**A**,**B**) refer to cut-off values for severe depression according to BDI (26 points) and PHQ9 (20 points).

**Table 1 nutrients-17-02573-t001:** Summary (median, 25–75%) of baseline characteristics for patients with super obesity (SMO) and non-super obesity (NSMO).

Variable	SMO N = 72	NSMO N = 247	*p*-Value
**Age [years]**	41 (34–50)	34 (29–43)	**<0.0001**
**Body mass (highest) [kg]**	154.5 (145–176.25)	128 (119–136.5)	**<0.0001**
**Body mass (present) [kg]**	149.5 (138.75–166.25)	122 (112–132)	**<0.0001**
**Height [cm]**	170 (165–180)	170 (165–177)	0.6067
**BMI (highest) [kg/m^2^]**	51.77 (50.93–54.59)	44.79 (42.14–46.92)	**<0.0001**
**BMI (present) [kg/m^2^]**	50.62 (49.57–53.2)	43.07 (40.45–45.28)	-

BMI—body mass index, SMO—super morbid obesity, NSMO—non-super morbid obesity.

**Table 2 nutrients-17-02573-t002:** Summary (N, %, odds ratio and 95% confidence interval) of socioeconomic data and history for patients with super morbid obesity (SMO) and non-super morbid obesity (NSMO).

Variable		SMO N = 72	NSMO N = 247	OR (95% CI)	*p*-Value
**Sex**	Male	19 (26.39%)	49 (19.84%)	1.45 (0.74–2.76)	0.2324
	Female ^R^	53 (73.61%)	198 (80.16%)		
**Employment**	Unemployed	26 (36.11%)	69 (27.94%)	1.46 (0.80–2.62)	0.1900
	Employed ^R^	46 (63.89%)	178 (72.06%)		
**Higher degree**	Yes	29 (40.28%)	86 (34.82%)	1.26 (0.71–2.23)	0.4056
	No ^R^	43 (59.72%)	161 (65.18%)		
**Relationship status**	Single	12 (16.67%)	46 (18.62%)	0.87 (0.40–1.81)	0.7048
	In relationship ^R^	60 (83.33%)	201 (81.38%)		
**Marital status**	Formal relationship/Marriage	40 (66.67%)	75 (37.31%)	3.36 (1.82–6.17)	0.0001
	Informal relationship	20 (33.33%)	126 (62.69%)		
**Lifelong obesity**	Yes	55 (76.39%)	102 (41.3%)	4.58 (2.45–8.92)	**<0.0001**
	No ^R^	17 (23.61%)	145 (58.7%)		
**Family history of obesity**	Yes	71 (98.61%)	217 (87.85%)	9.78 (1.57–405.33)	**0.0054**
	No ^R^	1 (1.39%)	30 (12.15%)		
**Higher degree of parents**	Yes	16 (22.22%)	101 (40.89%)	0.41 (0.21–0.78)	**0.0036**
	No ^R^	56 (77.78%)	146 (59.11%)		
**Violence against child**	Yes	36 (50%)	52 (21.05%)	3.73 (2.07–6.76)	**<0.0001**
	No ^R^	36 (50%)	195 (78.95%)		

^R^ Reference level, OR—odds ratio, 95% CI—95% confidence interval.

**Table 3 nutrients-17-02573-t003:** Summary (N, %, odds ratio with 95% confidence interval) for comorbidities in patients with super morbid obesity (SMO) and non-super morbid obesity (NSMO).

Variable		SMO N = 72	NSMO N = 247	OR (95% CI)	*p*-Value
**Heart attack**	Yes	4 (5.56%)	7 (2.83%)	2.01 (0.42–8.19)	0.2762
	No ^R^	68 (94.44%)	240 (97.17%)		
**Stroke**	Yes	3 (4.17%)	3 (1.21%)	3.52 (0.46–26.85)	0.1310
	No ^R^	69 (95.83%)	244 (98.79%)		
**Fatty liver disease**	Yes	69 (95.83%)	229 (92.71%)	1.81 (0.5–9.85)	0.4296
	No ^R^	3 (4.17%)	18 (7.29%)		
**Chronic venous insufficiency**	Yes	46 (63.89%)	4 (1.62%)	104.11 (34.34–428.85)	**<0.0001**
	No ^R^	26 (36.11%)	243 (98.38%)		
**Hypertension**	Yes	57 (79.17%)	128 (51.82%)	3.52 (1.85–7.07)	**0.0004**
	No ^R^	15 (20.83%)	119 (48.18%)		
**Obstructive sleep apnoea**	Yes	70 (97.22%)	147 (59.51%)	23.67 (6.06–203.12)	**<0.0001**
	No ^R^	2 (2.78%)	100 (40.49%)		
**Diabetes**	Yes	31 (43.06%)	107 (43.32%)	0.99 (0.56–1.74)	>0.9999
	No ^R^	41 (56.94%)	140 (56.68%)		

^R^ Reference level, OR—odds ratio, 95% CI—95% confidence interval.

**Table 4 nutrients-17-02573-t004:** Summary (median, 25–75%) of laboratory measurements for patients with super morbid obesity (SMO) and non-super morbid obesity (NSMO).

Variable	SMO N = 72	NSMO N = 247	*p*-Value
**AST [U/l]**	118.5 (100.75–134)	58 (48–76)	**<0.0001**
**ALT [U/l]**	121 (99–134)	68 (56–88)	**<0.0001**
**Total cholesterol [mg/dL]**	438.5 (348–562)	332 (268–402)	**<0.0001**
**HDL [mg/dL]**	39 (37–41)	42 (39–45)	**<0.0001**
**Triglycerides [mg/dL]**	268.5 (208–357.25)	211 (189–286)	**<0.0001**
**HbA1c [%]**	5.6 (5.5–6.53)	5.8 (5.6–5.9)	0.4313
**CRP [mg/dL]**	11.2 (7.8–14.3)	6.5 (4.9–11.15)	**<0.0001**
**Fasting glucose [mg/dL]**	98 (89.75–129)	100 (95–106.5)	0.5786
**Insulin [uU/mL]**	27.5 (23–40)	21 (17–26)	**<0.0001**
**HOMA-IR**	8.14 (5.38–10.38)	5.19 (3.95–7.01)	**<0.0001**

AST—aspartate transferase, ALT—alanine transaminase, HDL—high density lipoproteins, HbA1c—glycated hemoglobin, CRP—C-reactive protein, HOMA-IR—homeostatic model assessment insulin resistance.

**Table 5 nutrients-17-02573-t005:** Summary (N, %, odds ratio and 95% confidence interval) of psychiatric disease history and sleep parameters for patients with super morbid obesity (SMO) and non-super morbid obesity (NSMO).

Variable		SMO N = 72	NSMO N = 247	OR (95% CI)	*p*-Value
**History of psychiatric treatment**	Yes	15 (20.83%)	121 (48.99%)	0.28 (0.14–0.52)	**<0.0001**
	No ^R^	57 (79.17%)	126 (51.01%)		
**Current psychiatric treatment**	Yes	10 (13.89%)	57 (23.08%)	0.54 (0.23–1.15)	0.1019
	No ^R^	62 (86.11%)	190 (76.92%)		
**Depressive disorder**	Yes	13 (18.06%)	99 (40.08%)	0.33 (0.16–0.65)	**0.0010**
	No ^R^	59 (81.94%)	148 (59.92%)		
	No ^R^	70 (97.22%)	226 (91.5%)		
**Nocturnal eating syndrome**	Yes	30 (41.67%)	119 (48.18%)	0.77 (0.43–1.35)	0.3500
	No ^R^	42 (58.33%)	128 (51.82%)		
**Binge eating disorder**	Yes	29 (40.28%)	169 (68.42%)	0.31 (0.17–0.55)	**<0.0001**
	No ^R^	43 (59.72%)	78 (31.58%)		
**Bulimia**	Yes	10 (13.89%)	37 (14.98%)	0.92 (0.38–2.01)	>0.9999
	No ^R^	62 (86.11%)	210 (85.02%)		
**Number of eating disorders**	≥1	39 (54.17%)	199 (80,57%)	0.29 (0.16–0.50)	**<0.0001**
	<1	33 (45.83%)	48 (19.43%)		
**Subjective sleep quality**	Poor	50 (69.44%)	127 (51.42%)	2.14 (1.19–3.95)	**0.0071**
	Good ^R^	22 (30.56%)	120 (48.58%)		
**Prolonged sleep latency (>1 h)**	Yes	21 (29.17%)	61 (24.7%)	1.25 (0.66–2.32)	0.4469
	No ^R^	51 (70.83%)	186 (75.3%)		
**Nocturnal awakenings (>2/night)**	Yes	45 (62.5%)	122 (49.39%)	1.7 (0.97–3.05)	0.0604
	No ^R^	27 (37.5%)	125 (50.61%)		
**Early morning awakenings**	Yes	7 (9.72%)	34 (13.77%)	0.68 (0.24–1.64)	0.4289
	No ^R^	65 (90.28%)	213 (86.23%)		
**Snoring**	Yes	66 (91.67%)	118 (47.77%)	11.95 (4.95–35.02)	**<0.0001**
	No ^R^	6 (8.33%)	129 (52.23%)		
**Hypnotic medications**	Yes	1 (1.39%)	24 (9.72%)	0.13 (0.00–0.84)	**0.0222**
	No ^R^	71 (98.61%)	223 (90.28%)		

SMO—super morbid obesity, NSMO—non-super morbid obesity, ^R^ reference class.

## Data Availability

The data are not publicly available due to privacy and ethical restrictions.

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
