# Peer review of "Beyond Metabolism: Psychiatric and Social Dimensions in Bariatric Surgery Candidates with a BMI ≥ 50—A Prospective Cohort Study"

_nutrients, 2025, doi:10.3390/nu17152573_

Round 1
Reviewer 1 Report
Comments and Suggestions for Authors
I am a little concerned about the age difference here. I'm wondering if the older group is showing health outcome differences because of age. I'm thinking because of the metabolic strain of their condition on their organs that the longer lifespan is allowing these metabolic and physiological issues to be worse than the younger comparison group. I would recommend comparing similar age groups as a subset to see if those differences persist.
You mentioned in your conclusions that SMO patients should receive individualized bariatric treatment with an emphasis on mental health disorders. Do you think that would result in better post-op outcomes and if so why and what would that look like?
I'm not sure if this journal requires it but I would like to see some discussion about the limitations of this study and to the relative generalizability of your findings to other SMO patients.
I also don't see any discussion about how SMO patients in general fare after surgery. In other words, in general, do these patients experience worse than the post operation outcomes than the comparison group? Not specific to this study but overall. Because I think if that was the case you should mention it to further support your findings that they need more individualized care.
Author Response
First of all, thank you for your time and your essential and valuable comments regarding the article. Below I sent my responses.
Comments 1:
I am a little concerned about the age difference here. I'm wondering if the older group is showing health outcome differences because of age. I'm thinking because of the metabolic strain of their condition on their organs that the longer lifespan is allowing these metabolic and physiological issues to be worse than the younger comparison group. I would recommend comparing similar age groups as a subset to see if those differences persist.
Response 1:
Patients were classified as SMO when they achieved a BMI greater than 50. The consequences of the disease likely depend more on its duration than on the patient's age, and achieving a high BMI in older patients does not necessarily mean the duration of obesity; it often occurs in conjunction with important life-altering factors. We believe that achieving a BMI above 50 depends not only on the patient's age but also on the dynamics of obesity—"obesity acceleration"—and duration of obesity. Dividing the patient material into such subgroups is not only labor-intensive and methodologically challenging, but could also lead to the creation of numerous subgroups with small numbers of patients, which would distort the statistical methods used. The authors will attempt to group patients in this way as their number increases in our material, which is continually being updated in our database.
Comments 2:
You mentioned in your conclusions that SMO patients should receive individualized bariatric treatment with an emphasis on mental health disorders. Do you think that would result in better post-op outcomes and if so why and what would that look like?
Response 2:
Observations from numerous scientific studies indicate that the incidence of depressive disorders in patients with SMO is over 40%, and in most cases, patients do not receive psychiatric care. A very interesting observation is that patients, according to our psychometric scales (Beck, PHQ), which subjectively assess patients' depressiveness, meet the criteria for a depressive episode, often of moderate severity, yet do not seek psychiatric care. This is significantly influenced by environmental and metabolic factors, but above all, the duration of obesity, which often means these patients do not remember themselves as happy, motivated, or active. Another factor worsening the long-term outcome of the bariatric process is the underestimation of eating disorders in this group of patients, primarily compulsive eating disorder. Our observations indicate that incorporating antidepressant pharmacotherapy not only facilitates patient weight loss in the preoperative period, increases their sense of agency, and motivates them to undergo treatment, but also significantly improves compliance in the postoperative period. Patients in our study who were diagnosed with mental disorders, as well as those who had it previously, were included in the CBT psychotherapy process to develop better strategies for coping with difficult emotions, including behaviors related to compulsive eating in difficult situations.
Comments 3:
I'm not sure if this journal requires it but I would like to see some discussion about the limitations of this study and to the relative generalizability of your findings to other SMO patients.
Response 3:
A limitation of the study is the analysis of some variables based on non-standardized questionnaires. The ad hoc questionnaires were developed based on observations of therapists and physicians participating in follow-up visits, who often obtained information affecting patient functioning. This information, which could not be captured by well-known and widely used psychometric scales, would prevent the patient's condition, both pre- and post-operatively, from being captured. Expanding our patient interviews allowed us to gather a wealth of information that not only personalized the process for each patient but, we hope, will contribute to a long-term focus on factors that may indicate even better long-term outcomes and focus on factors associated with the superior success of the bariatric process.
Another limitation of the study: the attempt to create relatively homogeneous groups for comparison was limited to eliminating extreme results. We did not differentiate based on age, comorbidities, etc., which was due to the relatively small initial number of patients with SMO. We were concerned that introducing further subdivisions within the group would reduce the sample size and representativeness.
This is a multifaceted assessment, but the baseline assessment of patients with SMO is much more valuable. Follow-up of this group after bariatric treatment is much more valuable, and we expect to confirm the interesting observations that are being developed one and two years after the procedure. What seems valuable in this study is the comprehensive analysis of the characteristics of patients with SMO.
Our preliminary observations indicate that a comparison of patients with SMO and MO before the bariatric process and one and two years after surgery, due to their extensiveness, cannot be included in a single study.
Comments 4:
I also don't see any discussion about how SMO patients in general fare after surgery. In other words, in general, do these patients experience worse than the post operation outcomes than the comparison group? Not specific to this study but overall. Because I think if that was the case you should mention it to further support your findings that they need more individualized care.
Response 4:
The reviewer correctly noted that assessing the functioning of patients with SMO after surgery was not the focus of this study. The authors are currently analyzing the results of this group of patients at one and two years of follow-up, which will be the subject of a subsequent project.
I hope that the database being developed by statisticians will soon be the basis for a manuscript submitted for publication.
Numerous retrospective observations by the authors indicate that the improvement in functioning in many areas of life in patients with a BMI above 50 is significant, and additionally, bariatric and metabolic outcomes are satisfactory in a high percentage. However, this group fails to achieve the same results as patients with a BMI between 35 and 50. They are also significantly more likely to require further bariatric treatment. We will attempt to describe the reasons for this in a subsequent prospective study.
Patients with SMO reach this stage of obesity not only due to the duration of the disease but also due to delays in treatment resulting from other equally important factors, primarily psychogenic ones. Psychological and psychiatric support allows us to identify patient vulnerabilities and determine what additional therapeutic steps may improve long-term outcomes and enhance postoperative compliance in this group of patients. The results of the next phase of the study may indicate trends that may be included in a future manuscript.
Reviewer 2 Report
Comments and Suggestions for Authors
Thank you for the opportunity to review this important manuscript. Here are my comments and suggestions.
Check if the title should be ''Psychiatric and Social Dimensions in Bariatric Surgery Candidates with a BMI ≥50...''
Put ''victim of violence against children'' instead of ''childhood trauma'' in the abstract because it is a more precise term.
In the INTRODUCTION, the part ''In Poland, the most commonly per-61 formed bariatric procedures include laparoscopic sleeve gastrectomy (LSG), 62 Roux-en-Y gastric bypass (RYGB), and one-anastomosis gastric bypass (OAGB) 63 [9], with LSG recommended as the preferred initial step for super-obese patients 64 due to the elevated surgical risk [10]. Simultaneously, non-pharmacological treatments 65 should be implemented, emphasizing dietary modifications, increased physical activity, 66 and psychological and psychiatric support to enhance overall therapeutic success. To en-67 sure efficacy of those interventions, baseline psychosocial characteristics of surgery can-68 didates must be obtained in order to identify actionable areas.'' Should be removed. This is well known and not related to the topic. The only association that can be inferred is that the authors, in their discussion, identify the most effective surgical and other treatment modalities for bariatric surgery candidates with a BMI of 50 or higher, considering the psychiatric and social dimensions.
The authors state ''We did not observe any significant differences in unemployment rate (OR: 1.46, 135 95%CI: 0.80 - 2.62) between the study groups.'' We believe that the statistics are correct, but there is at least a tendency for unemployment to increase with higher BMI. That is logical. And physiological. But worrisome. This tendency is confirmed and described in other studies mentioned in the discussion section.
Author Response
Thank you in advance for such kind words according to the manuscript and for all valuable comments.
Comments 1:
Check if the title should be ''Psychiatric and Social Dimensions in Bariatric Surgery Candidates with a BMI ≥50...''
Response 1:
Thank you very much for your comment. The manuscript title has been corrected as suggested.
Comments 2:
Put ''victim of violence against children'' instead of ''childhood trauma'' in the abstract because it is a more precise term.
Response 2:
Thank you for pointing out the linguistic correctness. It has been changed as suggested.
Comments 3:
In the INTRODUCTION, the part ''In Poland, the most commonly per-61 formed bariatric procedures include laparoscopic sleeve gastrectomy (LSG), 62 Roux-en-Y gastric bypass (RYGB), and one-anastomosis gastric bypass (OAGB) 63 [9], with LSG recommended as the preferred initial step for super-obese patients 64 due to the elevated surgical risk [10]. Simultaneously, non-pharmacological treatments 65 should be implemented, emphasizing dietary modifications, increased physical activity, 66 and psychological and psychiatric support to enhance overall therapeutic success. To en-67 sure efficacy of those interventions, baseline psychosocial characteristics of surgery can-68 didates must be obtained in order to identify actionable areas.'' Should be removed. This is well known and not related to the topic. The only association that can be inferred is that the authors, in their discussion, identify the most effective surgical and other treatment modalities for bariatric surgery candidates with a BMI of 50 or higher, considering the psychiatric and social dimensions.
Response 3:
The commentary is very valuable. I agree that the mention of bariatric surgery in the introduction does not apply to the entire publication. This has been corrected as suggested.
Comments 4:
The authors state ''We did not observe any significant differences in unemployment rate (OR: 1.46, 135 95%CI: 0.80 - 2.62) between the study groups.'' We believe that the statistics are correct, but there is at least a tendency for unemployment to increase with higher BMI. That is logical. And physiological. But worrisome. This tendency is confirmed and described in other studies mentioned in the discussion section.
Response 4:
Thank you very much for bringing this issue to my attention, as it's a valid point. Although our study's results differed from those in most publications, it's a very valid point. It's logical that rising unemployment correlates with rising BMI. Furthermore, it's important to note that these patients primarily work remotely due to mobility issues, not to mention physical activity. An example of this is the study mentioned in the discussion, where J. O'Connell et al. noted the relationship between rising unemployment and increased body weight in the study group.
Reviewer 3 Report
Comments and Suggestions for Authors
This is a well-designed prospective cohort study examining the association between class IV obesity (BMI ≥50 Kg/m2) and various psychosocial parameters. The scientific quality of the work was assessed against the STROBE checklist (https://www.equator-network.org/reporting-guidelines/strobe/), which is appropriate for this study design (prospective cohort study). Most of the STROBE criteria were adequately addressed; however, several important points require clarification or revision:
1) Study size: The manuscript does not explicitly justify the study size, and I could not identify a power analysis. Please clarify whether the sample size was based on an a priori calculation, or explain if other factors (e.g., convenience sampling or feasibility) determined the cohort size.
2) Limitations & risk of bias: The Discussion section lacks a dedicated "limitations" section. You briefly mention "This observation may be attributed to a limitation of our study, namely the lack of data on antidiabetic medication use", referring to the self-reported prevalence of T2DM, but nothing beyond that. The manuscript would benefit from a more structured and comprehensive paragraph specifically addressing limitations and potential sources of bias, such as:
-- Selection bias due to the single-center design and specific referral patterns;
-- Recall bias from retrospective self-reporting in questionnaires (e.g., childhood trauma, psychiatric history);
-- Detection bias, as some psychiatric conditions were diagnosed post hoc during the study;
-- Attrition bias, related to the exclusion of 21 patients due to incomplete data.
While an exhaustive list is not necessary, it is important to highlight at least the main limitations of the study. Moreover, please expand on these points and consider how they might impact the generalizability and internal validity of your findings.
3) Statistical analysis: The manuscript would be strengthened by performing (or at least discussing the absence of) sensitivity analyses. Additionally, no adjusted multivariable models were used to control for potential confounders (e.g., age, sex, education). If such analyses were not feasible due to sample size or other constraints, please acknowledge this explicitly in the limitations section.
Additionally, the terminology used in the manuscript should be updated to align with current international standards. Specifically:
- The terms “super obesity” and “super obese” should be avoided. According to the IFSO-endorsed nomenclature (please see attachments), these terms are discouraged as they may contribute to obesity bias and stigmatization. Instead, use neutral and precise terms such as “class IV obesity” (BMI ≥50 Kg/m2) or “patients with BMI ≥50 Kg/m2”.
- Similarly, consider replacing the term “comorbidities” with “obesity-related health problems” to reflect the fact that obesity is a recognized disease in its own right, and to avoid language that implies secondary importance or causality.
In conclusion, this is a scientifically solid and clinically relevant study. With some refinements, particularly regarding limitations, terminology, and analytical transparency, it could make a valuable contribution to the literature. I look forward to the authors’ response to these suggestions.

Author Response
Thank you very much for such a thorough analysis of the manuscript and all your valuable and substantive comments.
Comments 1:
Study size: The manuscript does not explicitly justify the study size, and I could not identify a power analysis. Please clarify whether the sample size was based on an a priori calculation, or explain if other factors (e.g., convenience sampling or feasibility) determined the cohort size.
Response 1:
Thank you for the comment according to the study group. It was our main limitation to find proper number of patients with BMI ≥50 Kg/m2. The next limitation was to attempt to create relatively homogeneous groups for comparison. We did not differentiate based on age, comorbidities, etc., which was due to the relatively small initial. We were concerned that introducing further subdivisions within the group would reduce the sample size and representativeness.
Comments 2:
Limitations & risk of bias: The Discussion section lacks a dedicated "limitations" section. You briefly mention "This observation may be attributed to a limitation of our study, namely the lack of data on antidiabetic medication use", referring to the self-reported prevalence of T2DM, but nothing beyond that. The manuscript would benefit from a more structured and comprehensive paragraph specifically addressing limitations and potential sources of bias, such as: -- Selection bias due to the single-center design and specific referral patterns; - Recall bias from retrospective self-reporting in questionnaires (e.g., childhood trauma, psychiatric history); -- Detection bias, as some psychiatric conditions were diagnosed post hoc during the study; -- Attrition bias, related to the exclusion of 21 patients due to incomplete data. While an exhaustive list is not necessary, it is important to highlight at least the main limitations of the study. Moreover, please expand on these points and consider how they might impact the generalizability and internal validity of your findings.
Response 2:
This comment is very valuable, and I agree that a section on the study's limitations is missing. This has been corrected and incorporated into the manuscript. I agree that there are numerous limitations and errors related to the sample selection (as this is a single-center study), the retrospective patient assessment, and the detection of psychiatric disorders during the study. However, I would like to emphasize that this is an innovative, pilot study. We plan to expand the study group and eliminate as many bias as possible that could affect the results, so I would like to thank you for such a comprehensive and valuable comment.
Comments 3:
Statistical analysis: The manuscript would be strengthened by performing (or at least discussing the absence of) sensitivity analyses. Additionally, no adjusted multivariable models were used to control for potential confounders (e.g., age, sex, education). If such analyses were not feasible due to sample size or other constraints, please acknowledge this explicitly in the limitations section.
Response 3:
Thank you for the comment. The limitation according to statistical analysis was included in the section limitation. “The main limitation of the study was to attempt to create relatively homogeneous groups for comparison. We did not differentiate based on age, comorbidities, etc., which was due to the relatively small initial number of patients with SMO. We were concerned that introducing further subdivisions within the group would reduce the sample size and representativeness”.
Comments 4:
Additionally, the terminology used in the manuscript should be updated to align with current international standards. Specifically:
The terms “super obesity” and “super obese” should be avoided. According to the IFSO-endorsed nomenclature (please see attachments), these terms are discouraged as they may contribute to obesity bias and stigmatization. Instead, use neutral and precise terms such as “class IV obesity” (BMI ≥50 Kg/m2) or “patients with BMI ≥50 Kg/m2”.
Similarly, consider replacing the term “comorbidities” with “obesity-related health problems” to reflect the fact that obesity is a recognized disease in its own right, and to avoid language that implies secondary importance or causality.
Response 4:
All terminology has been corrected as suggested.